# Low Neutrophil-to-Lymphocyte Ratio Combined with High Intraepithelial CD8+ Tumour-Infiltrating Lymphocytes Within the Tumour Microenvironment Is a Prominent Prognostic Factor in Advanced Epithelial Ovarian Cancer

**DOI:** 10.3390/cancers17243904

**Published:** 2025-12-06

**Authors:** Mami Shibahara, Hiroshi Harada, Tomoko Kurita, Midori Murakami, Yoshikazu Harada, Toru Hachisuga, Shohei Shimajiri, Toshiyuki Nakayama, Yusuke Matsuura, Kiyoshi Yoshino

**Affiliations:** 1Department of Obstetrics and Gynaecology, School of Medicine, University of Occupational and Environmental Health, Kitakyushu 807-8555, Japan; shibahara-mami@med.uoeh-u.ac.jp (M.S.); t-kurita@med.uoeh-u.ac.jp (T.K.); m-midori@med.uoeh-u.ac.jp (M.M.); thachisu@med.uoeh-u.ac.jp (T.H.); k-yoshino@med.uoeh-u.ac.jp (K.Y.); 2Department of Pathology, School of Medicine, University of Occupational and Environmental Health, Kitakyushu 807-8555, Japan; harada-y@med.uoeh-u.ac.jp (Y.H.); shohei-s@med.uoeh-u.ac.jp (S.S.); toshi-n@med.uoeh-u.ac.jp (T.N.); 3Department of Nursing of Human Broad Development, University of Occupational and Environmental Health, Kitakyushu 807-8555, Japan; yusuke-m@med.uoeh-u.ac.jp

**Keywords:** neutrophil-to-lymphocyte ratio, tumour-infiltrating lymphocyte, epithelial ovarian cancer, cancer cachexia

## Abstract

Most patients with advanced epithelial ovarian cancer (EOC) exhibit a state resembling cancer cachexia due to elevated systemic inflammatory responses. Systemic inflammatory markers, such as the neutrophil-to-lymphocyte ratio (NLR), have been reported to correlate with poor prognosis. Tumour-infiltrating lymphocytes (TILs) significantly contribute to the prognosis of patients with EOC. Higher levels of cytotoxic T-cell infiltration in EOC are associated with significantly improved survival via cancer immunity. This study aimed to quantitatively assess TILs and NLR prior to the initial treatment of advanced EOC and to evaluate their prognostic value.

## 1. Introduction

Approximately 70% of patients with epithelial ovarian carcinoma (EOC) are at an advanced stage [1]. This is primarily because symptoms and findings are subjective and absent in the early stages, making diagnosis difficult [2,3]. Most patients with advanced EOC experience a state resembling cancer cachexia owing to elevated systemic inflammatory responses. Cancer cachexia is characterised by muscle wasting that does not fully improve with conventional nutritional support. It is triggered by systemic inflammation and increased catabolism, leading to malnutrition in patients with cancer, which negatively affects the prognosis and treatment outcomes [4,5]. Systemic inflammation influences prognosis in various cancers [6]. Even in EOCs, elevated systemic inflammatory markers, such as the neutrophil-to-lymphocyte ratio (NLR), have been reported to correlate with poor prognosis [7]. NLR is calculated by dividing the neutrophil count by the lymphocyte count, enabling the assessment of an immune system imbalance. It reflects shifts in the balance between innate and adaptive immunity, and evidence demonstrates its association with treatment outcomes in malignant tumours [8].

Recent studies have reported the impact of the tumour microenvironment (TME), particularly tumour-infiltrating lymphocytes (TILs), on the prognosis of various cancers [9]. Higher levels of cytotoxic T-cell infiltration in EOC are associated with significantly improved survival via cancer immunity [10,11]. However, research on TILs in patients with EOC other than high-grade serous carcinoma (HGSC) remains limited.

The prognostic impact of systemic inflammation and TILs has been reported in ovarian cancer (HGSC [12], endometrioid carcinoma, and clear cell carcinoma [13]). Nevertheless, the relationship between NLR and TILs in patients with EOC remains unclear. This study aimed to quantitatively assess TILs and NLR prior to the initial treatment of advanced EOC and to evaluate their prognostic value.

## 2. Materials and Methods

### 2.1. Patient Characteristics

We retrospectively reviewed 396 malignant ovarian tumours recorded in our database at the Department of Obstetrics and Gynaecology, University of Occupational and Environmental Health, Japan, between 2005 and 2020, and selected 101 patients diagnosed with EOC (stages III–IV according to the 2014 International Federation of Gynecology and Obstetrics (FIGO) classification). The patients had undergone adjuvant chemotherapy following primary debulking surgery (PDS), neoadjuvant chemotherapy (NACT)–interval debulking surgery (IDS), or first-line chemotherapy after staging laparoscopy or exploratory laparotomy as initial treatment. Standard therapies, including chemotherapy, poly (ADP-ribose) polymerase inhibitors (PARPi), or molecular targeted therapies such as bevacizumab, were administered as second-line or subsequent treatment following platinum-sensitive or -resistant recurrence. Patients who were unable to receive platinum-based chemotherapy as initial treatment, those with borderline ovarian malignant tumours, or those with complications from other cancers were excluded. Paraffin-embedded specimens were obtained from the University of Occupational and Environmental Health archives, and relevant clinical data were collected for analysis. The diagnostic criteria for EOC and the selection of stratification factors were based on the FIGO Cancer Report, as well as the recommendations from consensus meetings of the European Society of Gynaecological Oncology, the European Society for Medical Oncology, the European Society of Pathology, and the Gynaecological Cancer InterGroup [14,15,16]. The following factors were extracted from 101 patients: age at diagnosis, FIGO 2014 stage, histological type, initial platinum-based chemotherapy (NACT, adjuvant, or first-line), postoperative residual disease (R0, optimal surgery, suboptimal surgery, or exploratory laparotomy), NLR, albumin, ascites volume, TILs, progression-free survival (PFS), and overall survival (OS). PFS was defined as the interval from initial treatment to objective disease progression or final follow-up, with progression determined by imaging rather than by tumour marker elevation. OS was defined as the interval from the date of initial treatment to death or the last follow-up. NLR was calculated using clinical data obtained from venous blood samples collected within 4 weeks before initial treatment (surgery or chemotherapy) by dividing the absolute neutrophil count by the absolute lymphocyte count. Ascites grades were determined using computed tomography images as follows: grade 1, ascites confined to the upper abdomen or pelvic cavity; grade 2, ascites not meeting the criteria for grades 1 or 3; and grade 3, ascites extending continuously from the upper abdominal cavity to the pelvic cavity [17,18]. Two gynaecological oncology pathologists (M.S. and H.H.) independently reviewed the haematoxylin and eosin-stained slides and classified the histological subtypes of EOC. Any discrepancies in histopathological diagnosis were resolved through consultation between M.S. and H.H. For patients undergoing NACT, surgical biopsy specimens obtained before NACT initiation were analysed.

### 2.2. Immunohistochemistry

Formalin-fixed, paraffin-embedded tissue blocks obtained from ovarian cancer specimens were sectioned to a thickness of 4 μm. The sections were de-paraffinised and immunostained. The primary antibodies used were as follows: CD8 (monoclonal; catalogue no. #413951, dilution 1:2; Nichirei Bioscience, Tokyo, Japan), CD4 (monoclonal, catalogue no. #413201, dilution 1:2; Nichirei Bioscience, Tokyo, Japan), and programmed death-ligand 1 (PD-L1; monoclonal, catalogue no. #13684; dilution 1:200; Cell Signalling Technology, Danvers, MA, USA). Antigen retrieval was performed at 121 °C for 15 min in a 10 mM citrate buffer (pH 6.0). Secondary antibody binding was visualised using the Histofine Simple Stain MAX PO^®^ (Nichirei Bioscience, Tokyo, Japan). The slides were then counterstained with Meyer’s haematoxylin.

### 2.3. Digital Image Analysis

Immunohistochemical slides were digitally scanned at 20× magnification. Whole-slide images were analysed using HALO^®^ image analysis software (version 4.05, Indica Labs, Albuquerque, NM, USA). The tumours were classified into three distinct categories: intraepithelial, stromal, and non-contributory tissue. The density of each marker-positive cell was counted using HALO^®^ (Appendix A). Following a previous study [19], haematoxylin and eosin staining was confirmed, and the section with the highest TILs count in each EOC case was selected for analysis. The total number of CD8- and CD4-positive cells was expressed as density per mm^2^. The density of CD8+ and CD4+ ss in resected EOC specimens was measured in three patterns: combined intraepithelial and stromal (cTILs), intraepithelial alone (iTILs), and stromal alone (sTILs). The tumour proportion score (TPS) was calculated to determine PD-L1 expression [20]. TPS was defined as the proportion of surviving tumour cells showing partial or complete membrane staining of any intensity. Taking into account the specifications of the HALO image analysis software, analysis was performed using TPS. Based on the previous study [21], positive PD-L1 expression was defined as TPS ≥ 1%.

### 2.4. Statistical Analysis

Statistical analyses were performed using SPSS Statistics version 28 (IBM SPSS Statistics for Windows, IBM Corp., Armonk, NY, USA). Clinical and pathological data were evaluated using the chi-square test. Optimal cut-off values for NLR, albumin, and TILs (CD8+ and CD4+) were determined using the receiver operating characteristic (ROC) curve analysis in predicting death of the disease. The relationship between systemic inflammatory markers and TILs was assessed using Spearman’s correlation coefficients. OS and PFS were assessed using the Kaplan–Meier method and log-rank test. Univariate and multivariate Cox proportional hazards models were used to examine the association between potential risk factors and disease progression or death. Statistical significance was set at *p* < 0.05.

### 2.5. Ethical Review and Participant Consent

This study was approved by the Ethics Review Committee of the University of Occupational and Environmental Health Hospital (UOEHCRB21-155). This report does not present any identifiable images or other personal or clinical information that could compromise participant anonymity. The requirement for obtaining patient consent was waived owing to the retrospective design of the study and the use of de-identified data.

## 3. Results

### 3.1. EOC Characteristics

The characteristics of the 101 EOC cases are summarised in Table 1. The median age at diagnosis was 62 years (range, 34–82 years). FIGO stage III disease was present in 72 patients (72.3%), whereas stage IV disease was present in 29 patients (28.7%). The histological diagnoses were as follows: HGSC, 70 cases (69.3%); clear cell carcinoma, 19 cases (18.8%); endometrioid carcinoma, 9 cases (8.9%); and mucinous carcinoma, 3 cases (3.0%). Adjuvant chemotherapy was administered to 60 patients (59.6%), NACT to 29 patients (28.7%), and first-line chemotherapy to 12 patients (11.9%). The numbers of patients with ascites grades 1, 2, and 3 were 21 (20.8%), 34 (33.7%), and 46 (45.5%), respectively. The median observation period was 43.6 months (range, 1–181 months), with recurrence observed in 84 patients (83.2%) and death in 59 patients (58.4%).

### 3.2. Prognostic Value of Systemic Inflammatory Markers

Appendix A shows the median, range, and ROC curves of NLR and albumin. ROC analysis determined the cut-off values of 4.23 for NLR and 3.30 g/dL for albumin. Fifty-nine patients (58.4%) were in the low-NLR group, and 42 (41.6%) were in the high-NLR group. The low-albumin group comprised 40 patients (39.6%), while the high-albumin group consisted of 61 patients (60.3%). A strong negative correlation was observed between NLR and albumin levels (Spearman’s rho = −0.576, *p* < 0.001). No significant differences were observed in terms of postoperative residual tumours or staging laparotomies between the low- and high-NLR groups by chi-squared test (Table 2). The low-NLR group showed a tendency towards higher rates of adjuvant chemotherapy (*p* = 0.076). The high-NLR group showed a tendency towards stage IV disease, hypoalbuminaemia, and significantly more grade 3 ascites (*p* = 0.079, *p* = 0.095, and *p* < 0.001, respectively). Figure 1 shows the differences in OS and PFS according to NLR and albumin levels, as determined by the Kaplan–Meier method. The low-NLR and high-albumin level groups demonstrated significant prolongation of both OS and PFS (Figure 1a–d). The median OS was 93 months in the low-NLR group and 31 months in the high-NLR group (*p* < 0.001; Figure 1a). The median PFS in the low-NLR group was 20 months, compared with 12.5 months in the high-NLR group (*p* = 0.012; Figure 1b). The median OS in the high-albumin group was 97 months, compared with 32 months in the low-albumin group (*p* < 0.001; Figure 1c). The median PFS in the high-albumin group was 25 months, compared with 13 months in the low-albumin group (*p* = 0.003; Figure 1d).

### 3.3. Prognostic Value of TILs

Appendix A shows the median, range, and the ROC curves of CD8+/CD4+ cTILs, iTILs, and sTILs. ROC analysis identified the following cut-off values for CD8+/CD4+ TILs: 85.0/77.2 for cTILs, 92.9/194.7 for iTILs, and 63.4/119.5 for sTILs. Figure 2 shows the corresponding Kaplan–Meier curves for OS and PFS, comparing CD8+ cTILs, iTILs, and sTILs. A significant prolongation of both OS and PFS was observed in the groups with high levels of CD8+ TILs (Figure 2a–f). In the high-CD8+ cTIL group, the median OS was not reached (NR), whereas in the low-CD8+ cTIL group, it was 43 months, representing a significant difference (*p* = 0.002; Figure 2a). The median PFS was 18 months in the high-CD8+ cTIL group and 14 months in the low group (*p* = 0.033; Figure 2b). The median OS in the high-CD8+ iTIL group was 107 months, whereas a significant difference was observed at 39 months in the low-CD8+ iTIL group (*p* < 0.001; Figure 2c). The median PFS was 24.5 months in the high-CD8+ iTIL group and 13.0 months in the low-CD8+ iTIL group (*p* = 0.020; Figure 2d). The median OS in the high-CD8+ sTIL group was 100 months, compared with 44 months in the low-CD8+ sTIL group (*p* = 0.002; Figure 2e). The median PFS in the high-CD8+ sTIL group was 29 months, compared with 15 months in the low-CD8+ sTIL group (*p* = 0.001; Figure 2f). As shown in Table 3 (chi-squared test), no significant differences were observed in postoperative residual disease or staging laparotomies between the low- and high-CD8+ iTIL groups. The high-CD8+ iTIL group showed significantly more cases of HGSC histological type and NACT (*p* = 0.041 and 0.024, respectively). Appendix A shows the corresponding Kaplan–Meier curves for OS and PFS, comparing CD4+ cTILs, iTILs, and sTILs. Significant prolongation was observed in the groups with high CD4+ cTILs for both OS and PFS, and with high CD4+ sTILs for PFS.

### 3.4. Univariate and Multivariate Survival Analyses for OS and PFS

As shown in Table 4, the univariate analysis of OS revealed significant differences across eight categories. The multivariate analysis revealed significant differences in NLR (*p* = 0.015), albumin (*p* = 0.016), and CD8+ iTILs (*p* = 0.033). Furthermore, the univariate analysis of PFS showed significant differences across nine categories (Table 5). In contrast, the multivariate analysis revealed no significant differences, except in the residual disease category (R0 + optimal vs. suboptimal + staging laparoscopy; *p* = 0.028).

### 3.5. Independence of TILs and NLR as Prognostic Factors

The relationship between TILs and NLR was examined to evaluate whether NLR functions as an independent prognostic factor. NLR was negatively correlated with CD8+ cTILs, iTILs, and sTILs (Spearman’s rho = −0.0934, *p* = 0.353; rho = −0.0114, *p* = 0.91; rho = −0.147, *p* = 0.144, respectively), along with CD4+ cTILs, iTILs, and sTILs (rho = −0.0457, *p* = 0.65; rho = 0.0468, *p* = 0.642; rho = −0.0995, *p* = 0.322, respectively). In advanced EOC, NLR and CD8+ iTILs acted as independent prognostic indicators, showing no correlation. The 101 EOC cases were classified into four groups: low NLR and high CD8+ iTILs (N = 25), low NLR and low CD8+ iTILs (N = 16), high NLR and high CD8+ iTILs (N = 34), and high NLR and low CD8+ iTILs (N = 26). Analysis of the background factors by chi-squared test presented in Table 6 showed no significant intergroup differences, except for chemotherapy, albumin, and ascites (*p* = 0.001, *p* < 0.001, and *p* < 0.001). All 101 EOC cases were classified into and evaluated across four groups based on high or low NLR and high or low CD8+ iTILs. The 5-year OS rate was 82.2% (median survival time NR; range, 8–163 months) in the low-NLR group with high CD8+ iTILs (*n* = 25); 41.7% (46 months; range, 2–109 months) in the low-NLR group with low CD8+ iTILs (*n* = 16); 47.2% (52 months; range, 1–181 months) in the high-NLR group with high CD8+ iTILs (*n* = 34); and 26.0% (24 months; range, 2–106 months) in the high-NLR group with low CD8+ iTILs (*n* = 26). The 10-year OS rates were 61.6%, 31.2%, 19.3%, and NR, respectively. Figure 3a shows adjusted analyses including the covariates, chemotherapy, albumin, and ascites. In the low-NLR subgroup, OS was significantly prolonged in the high CD8+ iTILs group (*p* = 0.023; Figure 3b). Conversely, no significant difference was observed among the high NLR subgroups (*p* = 0.286; Figure 3c).

### 3.6. Evaluation of PD-L1 Expression (TPS)

We found that 78 of 101 patients (77.2%) had low PD-L1 expression (TPS  <  1%), while 23 patients (22.8%) had high PD-L1 expression (TPS  ≥  1%). Appendix A shows the differential Kaplan–Meier OS and PFS curves for high and low TPS values. No significantly prolonged survival was observed based on high or low TPS in either OS or PFS (*p* = 0.817 and *p* = 0.585, respectively).

### 3.7. Density of TILs and the Prognostic Value of TILs in HGSC and Non-HGSC

Appendix A shows the densities of CD8+ and CD4+ TILs in HGSC and other histological subtypes (clear cell, endometrioid, and mucinous carcinoma). The densities of CD8+ and CD4+ iTILs were significantly higher in HGSC than in other histological subtypes (clear cell, endometrioid, and mucinous carcinoma) (*p* = 0.025 and *p* = 0.002, respectively); however, no significant differences were observed in cTILs (*p* = 0.396 and *p* = 0.582, respectively) or sTILs (*p* = 0.208 and *p* = 0.108, respectively). In HGSC cases, OS was significantly prolonged in the groups with high levels of CD8+ cTILs, iTILs, and sTILs (*p* = 0.023, *p* = 0.004, and *p* = 0.013, respectively); however, PFS was not significantly prolonged (Appendix A). The multivariate analysis revealed significant differences in albumin (*p* = 0.004) and CD8+ iTILs (*p* = 0.032).

## 4. Discussion

This study investigated the prognostic significance of pre-treatment systemic inflammatory markers and TILs in patients with advanced EOC. NLR and CD8+ iTILs were identified as independent prognostic factors in this cohort for advanced EOC and showed no correlation. Therefore, the combined prognostic value of NLR and TILs was examined. Both NLR and TILs are associated with lymphocytes, which play key roles in acquired immunity, particularly in the immune defence responses that suppress the proliferation and migration of cancer cells. The suppression of cancer immunity has been reported to be a consequence of reduced blood lymphocyte counts due to cachectic tendencies or poor nutritional status [22]. Inflammation is essential for tissue repair, regeneration, and remodelling, and neutrophils are the key innate immune cells. The inflammatory response induces the accumulation of mutations and various epigenetic alterations in adjacent epithelial cells, thereby contributing to tumourigenesis [23]. They also contribute to malignant progression by producing inflammatory cytokines and chemokines, as well as secreting tumour growth factors such as vascular endothelial growth factor [24,25]. NLR serves as a biomarker for assessing immune system imbalance by linking two aspects of the immune system: the innate immune response mediated by neutrophils and the acquired immune response primarily supported by lymphocytes [26]. Meta-analyses have demonstrated that the median NLR cut-off is 4, and an NLR exceeding this cut-off independently predicts reduced OS across numerous tumours [6]. In this study, the optimal cut-off value for NLR in prognostic prediction was determined to be 4.23 based on the ROC curve (Appendix A). The current cut-offs are exploratory and cohort-specific. This aligns with the range (2.3–5.25) reported in previous ovarian cancer studies [7]. In this multivariate analysis, NLR was identified as a prognostic factor for OS but not for PFS. These findings are consistent with those reported in several other studies [27,28,29]. In cervical cancer, a low pre-treatment NLR is associated with a favourable treatment response to immune checkpoint inhibitors (ICIs) [30]. Our results suggest that NLR is an independent prognostic factor for advanced EOC and could serve as a biomarker of systemic cancer immune status.

CD8+ T cells recognise cancer-specific antigens from antigen-presenting cells, triggering local inflammation via cytokine release and mobilisation of secondary immune factors that eliminate cancer cells [31]. These processes constitute essential elements of antitumour immunity. Similarly to many other solid cancers, the presence of abundant CD8+ TILs within the cancer epithelium is a favourable prognostic indicator of ovarian cancer [10,32]. CD3 is expressed on almost all T cells, whereas CD8 and CD4 are expressed on cytotoxic and helper T cells, respectively [33,34]. However, definitive comparisons are difficult because most studies have performed only semi-quantitative assessments of TIL density [35]. In cytotoxic CD8+ T lymphocytes, PD-L1 suppresses effector function, leading to immune evasion and tumour proliferation [36]. A quantitative assessment was performed using image analysis software on slides subjected to immunohistochemical staining for CD4, CD8, and PD-L1 because of their close association with the cancer immune cycle. As in previous reports, CD8+ iTILs were independent prognostic factors [37,38]. In this study as well, the multivariate analysis showed that CD8+ iTILs were a prognostic factor and were selected as the primary TIL parameter. Furthermore, HGSC demonstrated a higher density of CD8-positive iTILs than did non-HGSCs. Previous studies have shown that HGSC exhibits significantly higher levels of CD8-positive TILs than do other EOCs [21]. The density of CD8+ iTILs in advanced EOC differed markedly between HGSC and other histological subtypes, suggesting that cut-off values may need to be considered separately for each histological type.

Although outcomes differ based on cancer type and study design, several studies have analysed the association between TILs and NLR in different cancers. In cases of gastric, colorectal, and triple-negative breast cancers, no correlation was observed between peripheral blood-based markers, including NLR and CD8+ TILs [39,40,41]. However, in non-small cell lung cancer, an inverse correlation was observed between CD3+ TILs and NLR [42]. Although NLR may reflect the immune status within the TME, no consistent conclusions have been reached regarding its relationship with TILs. In ovarian cancer, one study examined the prognostic significance of NLR and CD4+ and CD8+ TILs in HGSC cases but did not assess the relationship between NLR and TILs [12]. The present study found no correlation between NLR and TILs. Both were identified as independent prognostic factors in this cohort for advanced EOC in the multivariate analysis.

The 101 ovarian cancer cases were classified into four groups: high/low NLR and high/low CD8+ iTILs. We previously reported that TILs are independent favourable prognostic indicators for patients with HGSC undergoing suboptimal surgery [19]. The prognosis is particularly poor in cases of advanced EOC involving incomplete resection, defined as suboptimal surgery, where the largest residual tumour is 1 cm or larger [43]. However, background analysis confirmed minimal bias in prognostic factors (Table 2, Table 3, and Table 6) [19]. In the low-NLR and high-CD8+ iTIL group, despite the high proportion (40%) of suboptimal surgery cases, the 5- and 10-year survival rates were 82.2% and 61.6%, respectively, both significantly higher. Although no significant differences in residual disease were observed, the 5- and 10-year survival rates in the other three groups were considerably lower than those in the low-NLR and high-CD8+ iTIL group.

NLR is a blood marker that indicates an imbalance in the systemic immune system, specifically related to neutrophils and lymphocytes. In the context of cancer immunity, the lymphocyte-based immune system plays an important role [44,45]. As mentioned earlier, the impact of TILs on prognosis has been reported not only in ovarian cancer but also in other cancer types [32,38,46]. TILs reflect tumour-localised immunity. In particular, a higher number of CD8+ TILs adjacent to the tumour epithelium, rather than within the tumour stroma, is associated with a favourable prognosis [37,47]. In this study, CD8+ iTILs also functioned as independent prognostic factors in the multivariate analysis. The importance of cancer immunity is also evident from the recent widespread adoption of ICIs [48,49,50]. Two factors were identified as independent prognostic factors in our multivariate analysis: the predominance of lymphocytes in the circulating blood and the presence of cytotoxic TILs adjacent to the tumour. The group that included both factors had a superior long-term prognosis. This suggests that the level of cancer immunity before treatment may significantly affect prognosis. This finding is also relevant to the clinical question of whether reducing myelosuppression, an adverse event associated with treatment using cytotoxic anticancer agents, contributes to prolonged survival.

This study had some limitations. First, it was a retrospective study conducted at a single centre with a limited number of patients. Further accumulation of data with consistent histological types and treatment regimens is warranted. In particular, the number of non-HGSC cases was small. Second, the multivariate model in this study includes more than 10 covariates, which carries the risk of over-parametrisation. Third, differences in treatment approaches, such as chemotherapy, molecularly targeted agents, and PARPi, were not evaluated. The *BRCA* mutation and homologous recombination deficiency status were unknown. However, the study period spanned from 2005 to 2020, and the majority of patients with a favourable prognosis and an OS exceeding 5 years received only conventional therapy comprising surgery and platinum-based chemotherapy. The use of novel therapies, such as molecularly targeted drugs or PARPi, did not influence the long-term prognosis in this study. Although bevacizumab has been used since 2013, the GOG0218 trial involving bevacizumab in advanced ovarian cancer reported a 6-month survival benefit in the bevacizumab combination group compared with that in the non-combination group [51]. The survival curves for the low-NLR and high-CD8+ iTILs groups did not reach the median survival even after 10 years, indicating a significant prolongation of survival. This further indicates that NLR and iTILs function as independent prognostic factors. Fourth, a detailed elucidation of the molecular mechanisms directly related to this study has not yet been achieved. PD-L1 expression was not a prognostic indicator of advanced EOC (Appendix A). Several studies on HGSC of the ovary have reported conflicting results regarding PD-L1 expression. Some studies have indicated a favourable prognosis [52,53], whereas others have reported a poor prognosis [38] or no effect [54]. In this study, only four cases had TPS ≥ 5%, which was a small number; therefore, a cutoff value greater than 1% was not selected. Future analyses of other driver genes and proteins that specifically activate and enhance cancer immunity in patients with advanced EOC are warranted.

The combination of NLR and CD8+ iTILs has potential as a biomarker for long-term prognosis. Using NLR and CD8+ iTILs might be useful to identify treatment response to ICIs and other immunotherapies.

## 5. Conclusions

Low NLR combined with high CD8+ iTILs is a prominent prognostic factor in advanced EOC. The status of tumour-localised immunity and pre-treatment systemic inflammation had a significant impact on long-term prognosis.

## Figures and Tables

**Figure 1 cancers-17-03904-f001:**
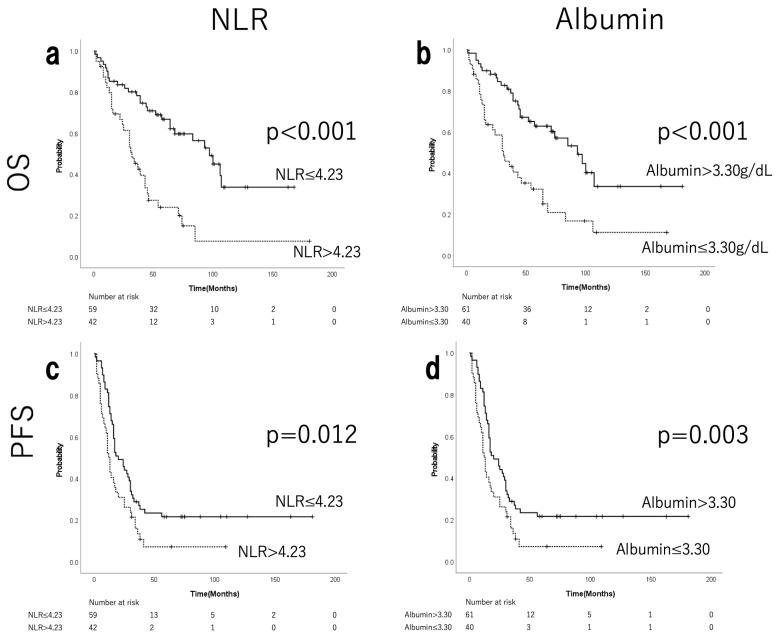
Kaplan–Meier curves for OS and PFS of patients with EOC stratified by pre-treatment NLR (**a**,**c**) and albumin (**b**,**d**). Patients with NLR  ≤ 4.23 and albumin levels >3.30 g/dL were significantly associated with better OS and PFS than those with NLR  > 4.23 and albumin levels  ≤3.30 g/dL. NLR, neutrophil-to-lymphocyte ratio; OS, overall survival; PFS, progression-free survival; EOC, epithelial ovarian cancer.

**Figure 2 cancers-17-03904-f002:**
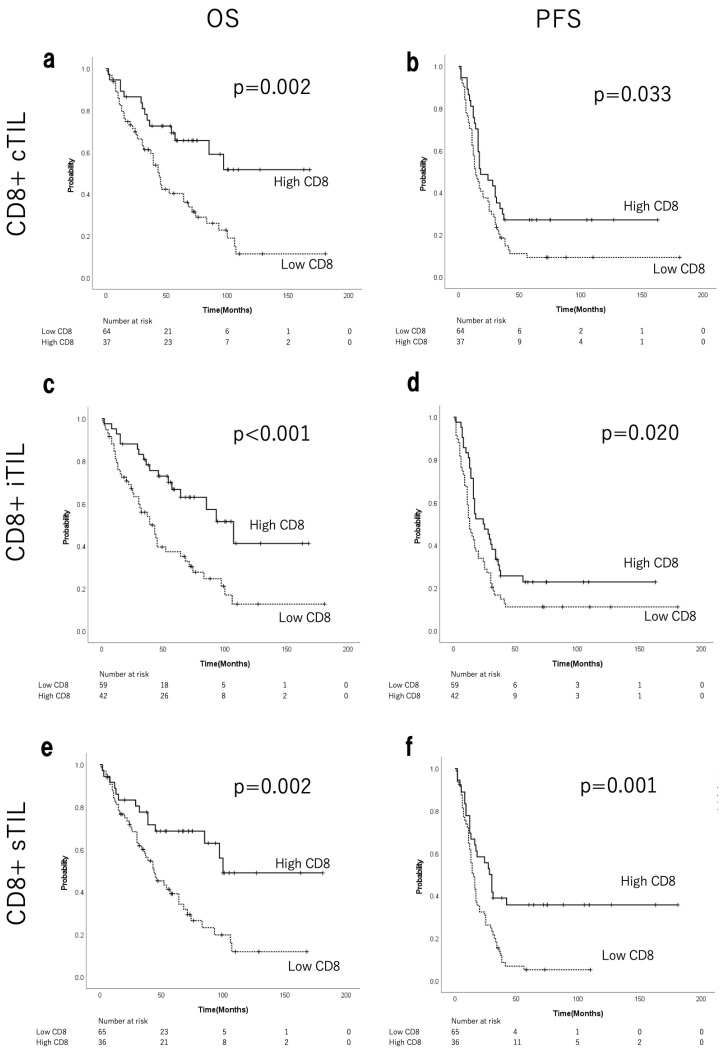
Kaplan–Meier curves for OS and PFS of patients with EOC stratified by CD8+ cTILs (**a**,**b**), iTILs (**c**,**d**), and sTILs (**e**,**f**). OS, overall survival; PFS, progression-free survival; EOC, epithelial ovarian cancer; TILs, tumour-infiltrating lymphocytes; c, combined (intraepithelial + stromal); i, intraepithelial; s, stromal.

**Figure 3 cancers-17-03904-f003:**
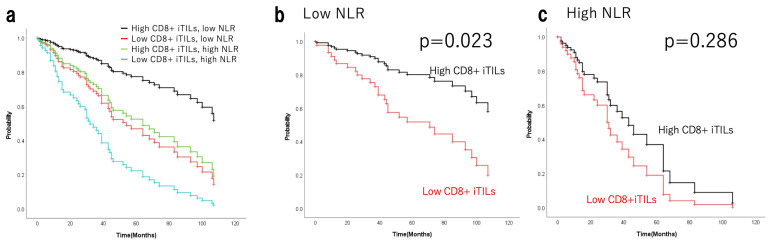
Survival curves adjusted analyses including covariates for OS of patients with epithelial ovarian cancer based on high/low NLR with/without CD8+ iTILs (**a**) and stratified by CD8+ iTILs in the low-NLR (**b**) and high-NLR (**c**) groups. OS, overall survival; TILs, tumour-infiltrating lymphocytes; i, intraepithelial; NLR, neutrophil-to-lymphocyte ratio.

**Table 1 cancers-17-03904-t001:** Clinicopathological parameters.

N = 101		N (%)
Age (years)	<50	23 (22.8%)
	≥50	78 (77.2%)
Stage	Ⅲ	72 (72.3%)
	Ⅳ	29 (28.7%)
Lymph node metastasis	Yes	29 (28.7%)
	No	29 (28.7%)
	Unknown	43 (42.6%)
Chemotherapy	NACT	29 (28.7%)
	Adjuvant	60 (59.4%)
	First-line	12 (11.9%)
Residual disease	Complete (R = 0)	20 (19.8%)
	Optimal (≤1 cm)	38 (37.6%)
	Suboptimal (>1 cm)	31 (30.7%)
	Exploratory laparotomy	12 (11.9%)
Histological subtype	HGSC	70 (69.3%)
	Clear cell	19 (18.8%)
	Endometrioid	9 (8.9%)
	Mucinous	3 (3.0%)
NLR	≤4.23	59 (58.4%)
	>4.23	42 (41.6%)
Albumin	≤3.30 g/dL	40 (39.6%)
	>3.30 g/dL	61 (60.3%)
Ascites	Grade 1	21 (20.8%)
	Grade 2	34 (33.7%)
	Grade 3	46 (45.5%)
Recurrence	Yes	84 (83.2%)
	No	17 (16.8%)
Prognosis	NED + AWD	42 (41.6%)
	DOD	59 (58.4%)

NLR, neutrophil-to-lymphocyte ratio; NACT, Neo-adjuvant chemotherapy; HGSC, high-grade serous carcinoma; NED, no evidence of disease; AWD, alive with disease; DOD, dead with disease.

**Table 2 cancers-17-03904-t002:** Correlation between clinical and biological parameters with pre-treatment NLR.

Clinicopathological Association with NLR		NLR ≤ 4.23 (N = 59)	NLR > 4.23 (N = 42)	*p*-Value
Age	<50	13	10	0.834
	≥50	46	32	
Stage	Ⅲ	46	26	0.079
	Ⅳ	13	16	
Histological subtype	HGSC	44	26	0.355
	Clear cell	10	9	
	Endometrioid	3	6	
	Mucinous	2	1	
Lymph node metastasis	Yes	21	8	0.192
	No	15	14	
Chemotherapy	NACT	12	17	0.076
	Adjuvant	40	20	
	First-line	7	5	
Residual disease	Complete (R = 0)	13	7	0.913
	Optimal (≤1 cm)	22	16	
	Suboptimal (>1 cm)	17	14	
	Exploratory laparotomy	7	5	
Albumin	≤3.30 g/dL	14	26	0.095
	>3.30 g/dL	45	16	
Ascites	Grade 1	17	4	<0.001 ^§^
	Grade 2	25	9	
	Grade 3	17	29	

NLR, neutrophil-to-lymphocyte ratio; HGSC, high-grade serous carcinoma; NACT, neo-adjuvant chemotherapy; ^§^ < 0.001.

**Table 3 cancers-17-03904-t003:** Correlation between clinical and biological parameters with CD8+ iTILs.

Clinicopathological Association with CD8+ iTILs		Low CD8+ iTILs	High CD8+ iTILs	*p*-Value
Age	<50	14	9	0.81
	≥50	45	33	
Stage	Ⅲ	42	30	1
	Ⅳ	17	12	
Histological subtype	HGSC	35	35	0.041 *
	Clear cell	16	3	
	Endometrioid	5	4	
	Mucinous	3	0	
Lymph node metastasis	Yes	17	12	0.43
	No	13	16	
Chemotherapy	NACT	11	18	0.024 *
	Adjuvant	39	21	
	First-line	9	3	
Residual disease	Complete (R = 0)	10	10	0.245
	Optimal (≤1 cm)	19	19	
	Suboptimal (>1 cm)	21	10	
	Exploratory laparotomy	9	3	
NLR	≤4.23	33	26	0.68
	>4.23	26	16	
Albumin	≤3.30 g/dL	27	13	0.097
	>3.30 g/dL	32	29	
Ascites	Grade 1	9	12	0.13
	Grade 2	24	10	
	Grade 3	26	20	

NLR, neutrophil-to-lymphocyte ratio; TILs, tumour-infiltrating lymphocytes; NACT, neo-adjuvant chemotherapy. * *p* < 0.05.

**Table 4 cancers-17-03904-t004:** Univariate and multivariate survival time analyses for overall survival (OS) of the 101 patients with epithelial ovarian cancer.

		Univariate Analysis		Multivariate Analysis	
		HR (95% CI)	*p*-Value	HR (95% CI)	*p*-Value
Age	50> vs. ≥50	0.971 (0.523–1.803)	0.927		
Stage	Ⅲ vs. Ⅳ	0.679 (0.397–1.159)	0.156		
Lymph node metastasis	Yes vs. No	0.743 (0.357–1.549)	0.420		
Chemotherapy	NACT vs. adjuvant + first-line	0.979 (0.554–1.731)	0.941		
Residual disease	Complete + optimal vs. Suboptimal + exploratory laparotomy	0.566 (0.339–0.947)	0.030 *	0.699 (0.402–1.216)	0.205
Albumin	3.30≥ vs. >3.30	3.051 (1.792–5.196)	<0.001 §	2.183 (1.166–4.085)	0.015 *
NLR	4.23≥ vs. >4.23	0.383 (0.228–0.642)	<0.001 §	0.486 (0.270–0.875)	0.016 *
Ascites	Grade 1 + 2 vs. Grade 3	0.413 (0.243–0.702)	0.001 †	0.771 (0.405–1.469)	0.429
CD8+ cTILs	High vs. low	0.394 (0.216–0.720)	0.002 †	1.360 (0.538–3.440)	0.516
CD8+ iTILs	High vs. low	0.391 (0.222–0.690)	0.001 †	0.430 (0.199–0.933)	0.033 *
CD8+ sTILs	High vs. low	0.389 (0.212–0.714)	0.002 †	0.599 (0.279–1.283)	0.187
CD4+ cTILs	High vs. low	0.335 (0.191–0.588)	<0.001 §	0.592 (0.313–1.121)	0.108
CD4+ iTILs	High vs. low	0.619 (0.366–1.045)	0.073		
CD4+ sTILs	High vs. low	0.616 (0.360–1.054)	0.077		
TPS	≥1% vs. <1%	0.928 (0.491–1.753)	0.818		

HR, hazard ratio; CI, confidence interval; NACT, Neo-adjuvant chemotherapy; TILs, tumour-infiltrating lymphocytes; c, combined (intraepithelial + stromal); i, intraepithelial; s, stromal; TPS, tumour proportion score. * *p* < 0.05, † *p* < 0.005, § *p* < 0.001. Cases with ‘unknown’ lymph node status were excluded from the analysis.

**Table 5 cancers-17-03904-t005:** Univariate and multivariate survival time analyses for progression-free survival (PFS) of the 101 patients with epithelial ovarian cancer.

		Univariate Analysis		Multivariate Analysis	
		HR (95% CI)	*p*-Value	HR (95% CI)	*p*-Value
Age	50> vs. ≥50	0.745 (0.431–1.288)	0.292		
Stage	Ⅲ vs. Ⅳ	0.676 (0.424–1.077)	0.100		
Lymph node metastasis	Yes vs. No	0.744 (0.357–1.549)	0.570		
Chemotherapy	NACT vs. adjuvant + first-line	1.097 (0.693–1.737)	0.694		
Residual disease	Complete + optimal vs. Suboptimal + exploratory laparotomy	0.517 (0.335–0.798)	0.003 †	0.584 (0.362–0.943)	0.028 *
Albumin	3.30≥ vs. >3.30	1.908 (1.231–2.959)	0.004 †	1.400 (0.836–2.346)	0.201
NLR	4.23≥ vs. >4.23	0.585 (0.379–0.901)	0.015 *	0.698 (0.421–1.159)	0.165
Ascites	Grade 1 + 2 vs. Grade 3	0.558 (0.362–0.861)	0.008 *	0.862 (0.498–1.491)	0.594
CD8+ cTILs	High vs. low	0.615 (0.389–0.975)	0.039 *	1.493 (0.714–3.124)	0.287
CD8+ iTILs	High vs. low	0.601 (0.386–0.935)	0.024 *	0.549 (0.282–1.068)	0.077
CD8+ sTILs	High vs. low	0.462 (0.284–0.753)	0.002 †	0.592 (0.330–1.060)	0.078
CD4+ cTILs	High vs. low	0.478 (0.298–0.767)	0.002 †	0.838 (0.452–1.554)	0.575
CD4+ iTILs	High vs. low	0.871 (0.567–1.338)	0.528		
CD4+ sTILs	High vs. low	0.536 (0.339–0.847)	0.008 *	0.726 (0.406–1.296)	0.297
TPS	≥1% vs. <1%	0.865 (0.507–1.473)	0.593		

HR, hazard ratio; CI, confidence interval; NACT, neo-adjuvant chemotherapy; TILs, tumour-infiltrating lymphocytes; c, combined (intraepithelial + stromal); i, intraepithelial; s, stromal; TPS, tumour proportion score. * *p* < 0.05, † *p* < 0.005. Cases with ‘unknown’ lymph node status were excluded from the analysis.

**Table 6 cancers-17-03904-t006:** Correlation of clinical and biological parameters with pre-treatment NLR and iCD8+ TILs.

		NLR < 4.23		NLR ≥ 4.23		
		High CD8+ iTILs (N = 25)	Low CD8+ iTILs (N = 16)	High CD8+ iTILs (N = 34)	Low CD8+ iTILs (N = 26)	*p*-Value
Age	<50	5	4	8	6	0.983
	≥50	20	12	26	20	
Stage	Ⅲ	18	11	28	15	0.218
	Ⅳ	7	5	6	11	
Histological subtype	HGSC	22	12	24	16	0.553
	Clear cell	2	1	7	5	
	Endometrioid	1	2	2	4	
	Mucinous	0	1	1	1	
Lymph node metastasis	Yes	7	8	8	6	0.267
	No	9	3	12	5	
	Unknown	9	5	14	15	
Chemotherapy	NACT	6	12	6	5	0.001 †
	Adjuvant	16	4	24	16	
	First-line	3	0	4	5	
Residual disease	Complete (R = 0)	5	5	8	2	0.407
	Optimal (≤1 cm)	10	8	12	8	
	Suboptimal (>1 cm)	7	3	10	11	
	Exploratory laparotomy	3	0	4	5	
Albumin	≤3.30 g/dL	4	9	10	17	<0.001 §
	>3.30 g/dL	21	7	24	9	
Ascites	Grade 1	11	1	6	3	<0.001 §
	Grade 2	8	2	17	7	
	Grade 3	6	13	11	16	

TILs, tumour-infiltrating lymphocytes; i, intraepithelial; NLR, neutrophil-to-lymphocyte ratio; NACT, neo-adjuvant chemotherapy; HGSC, high-grade serous carcinoma. † *p* < 0.005, § *p* < 0.001.

## Data Availability

Data presented in this study are available upon request from the corresponding author. The data are not publicly available due to privacy and ethical restrictions.

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
