# Peer review of "Cancers2025, 17(24), 3904;https://doi.org/10.3390/cancers17243904"

_cancers, 2025, doi:10.3390/cancers17243904_

Round 1

Reviewer 1 Report

Comments and Suggestions for Authors

In this manuscript, authors explored that the relationship between systemic inflammatory marker and tumor infiltrating lymphocyte with prognosis of patients suffering from epithelial ovarian cancer. Their investigation revealed that a low neutrophil-to-lymphocyte ratio and high intraepithelial CD8+ tumor-infiltrating lymphocytes served as independent prognostic factor in advanced epithelial ovarian cancer. Nevertheless, several limitations were identified within the study. From my perspective, certain significant revisions need to be addressed prior to acceptance for publication.

  1. Title and abstract require enhancement. Although the authors examined the associations of CD4+, CD8+, and PD-L1+ expressions with overall survival (OS) and progression-free survival (PFS) in a cohort of 101 epithelial ovarian cancer (EOC) patients, they omitted detailed descriptions beyond CD8+ lymphocytes in the abstract.
  2. The major findings are not adequately emphasized. It is suggested that sections 3.2/3.3 and 3.4/3.5 be consolidated to refine the narrative flow and strengthen the presentation of core conclusions.
  3. Figure S1 solely illustrates an example of digital Tumor-Infiltrating Lymphocyte (TIL) quantification using the HALO® image analysis platform. It is advisable to include representative images depicting the outcomes of immunohistochemical analyses for CD4+, CD8+, and PD-L1+ expressions.
  4. All tables should adhere to the three-line table format. Specifically, in Table 2, while the p-value is indicated as <0.001*, the corresponding note references *p<0.05, which requires clarification or correction.
  5. The conclusion lacks explicit articulation of the study's principal findings and viewpoints. A more comprehensive summary highlighting these aspects is necessary.

Author Response

In this manuscript, authors explored that the relationship between systemic inflammatory marker and tumor infiltrating lymphocyte with prognosis of patients suffering from epithelial ovarian cancer. Their investigation revealed that a low neutrophil-to-lymphocyte ratio and high intraepithelial CD8+ tumor-infiltrating lymphocytes served as independent prognostic factor in advanced epithelial ovarian cancer. Nevertheless, several limitations were identified within the study. From my perspective, certain significant revisions need to be addressed prior to acceptance for publication.

-To Reviewer 1

We are most grateful for your peer review of the manuscript and for your valuable suggestions. The amended sections have been highlighted with yellow lines.

Title and abstract require enhancement. Although the authors examined the associations of CD4+, CD8+, and PD-L1+ expressions with overall survival (OS) and progression-free survival (PFS) in a cohort of 101 epithelial ovarian cancer (EOC) patients, they omitted detailed descriptions beyond CD8+ lymphocytes in the abstract.

-We have revised the title. The relevant section of the abstract has been updated, and descriptions of CD4 and PD-L1 expression have been added (lines 35–37).

The major findings are not adequately emphasized. It is suggested that sections 3.2/3.3 and 3.4/3.5 be consolidated to refine the narrative flow and strengthen the presentation of core conclusions.

-Sections 3.2 and 3.3, as well as 3.4 and 3.5, have been merged, and the section numbers have been updated. Additionally, it is explicitly stated that the median and range for each dataset can be found in Supplementary Figures S2 and S3.

Figure S1 solely illustrates an example of digital Tumor-Infiltrating Lymphocyte (TIL) quantification using the HALO® image analysis platform. It is advisable to include representative images depicting the outcomes of immunohistochemical analyses for CD4+, CD8+, and PD-L1+ expressions.

-As you pointed out, we have added images of immunohistochemical analyses of CD4+, CD8+, and PD-L1+ expression to Figure S1.

All tables should adhere to the three-line table format. Specifically, in Table 2, while the p-value is indicated as <0.001*, the corresponding note references *p<0.05, which requires clarification or correction.

-As you pointed out, I have applied the three-line table format to all of the tables. We defined the p-value as follows: *p < 0.05, †p < 0.005, and §p < 0.001.

The conclusion lacks explicit articulation of the study's principal findings and viewpoints. A more comprehensive summary highlighting these aspects is necessary.

-We have revised the Conclusion. Thank you for your valuable suggestions.

Reviewer 2 Report

Comments and Suggestions for Authors

This manuscript addresses a relevant topic: the combined prognostic impact of pre‑treatment NLR and TILs in advanced EOC, using digital image analysis. However, several numerical inconsistencies and statistical issues need to be addressed before the conclusions can be considered robust.

1. Numerical and labelling inconsistencies

1.1 Median PFS according to NLR (Section 3.3 / Figure 1b)

The median PFS is reported as 93 months in the low‑NLR group and 31 months in the high‑NLR group, which is not plausible for stage III–IV EOC and likely reflects a copy of the OS values. Please recalculate and correct the median PFS so that it matches Figure 1b.

1.2 Albumin distribution: Table 1 vs Table 2

The numbers of patients with low vs high albumin are reversed between Table 1 and Table 2, and the text statement (“high‑NLR group tended to have low albumin”) does not match the table. Please reconcile the albumin categories and counts across tables and text.

1.3 Cox model hazard ratios and reference categories (Table 4)

In the multivariate OS model, high albumin and low NLR are associated with HR > 1 (“>3.3 vs ≤3.3”, “≤4.23 vs >4.23”), which contradicts the Kaplan–Meier curves where these groups have better survival. Similar issues appear for residual disease and ascites. Please re‑check reference coding and labels for all variables in Table 4 and correct HRs/wording accordingly.

1.4 Correlation between NLR and CD8+ iTILs (Section 3.7)

A Spearman’s rho of −0.353 with p = 0.91 is implausible; this suggests a reporting error, yet you conclude that NLR and TILs are uncorrelated. Please re‑calculate rho and p‑values and revise the interpretation if needed.

1.5 Typos and minor labels

Examples include “Clinicopathological association with eCD8+ TIL” in Table 3 and “Renge” in Supplementary Figure S2. Please correct these throughout.

2. Statistical analysis

2.1 ROC‑derived cut‑offs and dichotomisation

NLR, albumin and all six TIL indices use “optimal” cut‑offs derived from ROC analysis in the same cohort, which are then used to dichotomise the same data for KM/Cox analyses. This approach risks overfitting and optimistic HRs. If possible, please consider modelling NLR/TILs as continuous variables or using pre‑specified cut‑offs, and clearly state in the Discussion that the current cut‑offs are exploratory and cohort‑specific.

2.2 Number of covariates vs events

There are 59 deaths for OS and 84 events for PFS, yet more than 10 covariates are included in the multivariable models. Please consider a more parsimonious model (fewer key covariates or penalised methods) and/or acknowledge the risk of over‑parameterisation as a limitation.

2.3 Description of statistical tests

The Methods state that clinical/pathological data were analysed using Mann–Whitney U tests, but Tables 2–3 compare categorical variables where chi‑square or Fisher’s exact tests would typically be used. Please specify which test was actually used for each table and re‑analyse if necessary.

2.4 Handling of missing lymph node status

Lymph node status is “Unknown” in 43/101 patients, but Cox models report only “Yes vs No.” Please clarify how “Unknown” was handled (exclusion, separate category, etc.) and its potential impact.

2.5 Confounding in the four‑group NLR × CD8+ iTIL analysis

Table 5 shows strong imbalances in chemotherapy regimen, albumin and ascites across the four groups (all p < 0.001), yet Figure 3 compares survival without adjustment. Please provide adjusted analyses including these covariates, or clearly discuss confounding as a limitation.

3. Interpretation and discussion

3.1 “Independent prognostic factor”

Given the data‑driven cut‑offs, complex models and correlation issues, the term “independent prognostic factor” for NLR and CD8+ iTIL may be somewhat strong. I suggest softening the wording (e.g. “independent in this cohort”) and expanding the discussion of statistical limitations and the need for validation.

3.2 Choice of CD8+ iTIL as main TIL marker

Figure S3 shows the highest AUC for CD4+ cTIL, yet CD8+ iTIL is used as the central marker. Please briefly explain why CD8+ iTIL was selected as the primary TIL parameter (biological rationale, multivariable results, prior literature).

3.3 Histology (HGSC vs non‑HGSC)

Figure S6 demonstrates higher CD8+/CD4+ iTIL density in HGSC vs non‑HGSC, while histology is not included as a covariate in the Cox models. Please consider including histology in multivariable analyses or provide HGSC‑only multivariable results, and highlight histology‑dependence of TIL cut‑offs as a limitation.

3.4 PD‑L1 (TPS) evaluation

PD‑L1 is dichotomised using TPS ≥ 1% on tumour cells only; the rationale and limitations are only briefly mentioned. A short justification for the 1% cut‑off and a note on not assessing immune‑cell PD‑L1 would strengthen the discussion.

4. Minor presentation issues

4.1 “Number at risk” in Kaplan–Meier plots

Please add numbers at risk under all KM curves (Figures 1–3 and supplementary KM figures), which will improve interpretability.

4.2 Consistency of terminology for chemotherapy regimens

Terms such as “Adjuvant,” “Neo‑adjuvant,” “Initial,” “First‑line,” and “NACT‑IDS” are used inconsistently across text and tables. Please define each in the Methods and harmonise terminology.

4.3 Abbreviations and legends

Abbreviations (c/i/sTILs, TPS, etc.) are not always defined in the figure legends or are used with minor variations. Please standardise abbreviations and ensure that each legend defines all terms used.

I hope these concise comments are helpful in revising the manuscript and strengthening the validity and clarity of your findings.

< !-- notionvc: 5c408241-d503-4de9-a376-c570336bc03d -->

Author Response

This manuscript addresses a relevant topic: the combined prognostic impact of pre‑treatment NLR and TILs in advanced EOC, using digital image analysis. However, several numerical inconsistencies and statistical issues need to be addressed before the conclusions can be considered robust.

-To Reviewer 2

-We appreciate your review of the manuscript and your valuable suggestions. We are also grateful for your identification of the transcription errors. The amended sections have been highlighted with yellow lines.

  1. Numerical and labelling inconsistencies

1.1 Median PFS according to NLR (Section 3.3 / Figure 1b)

The median PFS is reported as 93 months in the low‑NLR group and 31 months in the high‑NLR group, which is not plausible for stage III–IV EOC and likely reflects a copy of the OS values. Please recalculate and correct the median PFS so that it matches Figure 1b.

-We have corrected the median PFS to 20 months in the low NLR group and 12.5 months in the high NLR group (lines 204,205).

1.2 Albumin distribution: Table 1 vs Table 2

The numbers of patients with low vs high albumin are reversed between Table 1 and Table 2, and the text statement (“high‑NLR group tended to have low albumin”) does not match the table. Please reconcile the albumin categories and counts across tables and text.

-As you pointed out, the number of patients with low versus high albumin levels in Tables 1 and 2 (and also in Table 3) was incorrect. We revised the numbers to 40 patients with low albumin and 61 patients with high albumin in Tables 1-3.

1.3 Cox model hazard ratios and reference categories (Table 4)

In the multivariate OS model, high albumin and low NLR are associated with HR > 1 (“>3.3 vs ≤3.3”, “≤4.23 vs >4.23”), which contradicts the Kaplan–Meier curves where these groups have better survival. Similar issues appear for residual disease and ascites. Please re‑check reference coding and labels for all variables in Table 4 and correct HRs/wording accordingly.

-As you pointed out, there were some errors in the labels of the univariate and multivariate analyses. We revised the labels for age, stage, lymph node metastasis, residual disease, albumin, NLR, and ascites in Tables 4 and 5 for both analyses.

1.4 Correlation between NLR and CD8+ iTILs (Section 3.7)

A Spearman’s rho of −0.353 with p = 0.91 is implausible; this suggests a reporting error, yet you conclude that NLR and TILs are uncorrelated. Please re‑calculate rho and p‑values and revise the interpretation if needed.

-We sincerely apologize for the incorrect statement regarding Spearman’s rho. It has been corrected to rho = -0.0114 with p = 0.91 (line 284/ Section new 3.5).

1.5 Typos and minor labels

Examples include “Clinicopathological association with eCD8+ TIL” in Table 3 and “Renge” in Supplementary Figure S2. Please correct these throughout.

-We apologize for the typographical errors. The typos in Table 3 and Figures S2 and S3 have been corrected.

  1. Statistical analysis

2.1 ROC‑derived cut‑offs and dichotomisation

NLR, albumin and all six TIL indices use “optimal” cut‑offs derived from ROC analysis in the same cohort, which are then used to dichotomise the same data for KM/Cox analyses. This approach risks overfitting and optimistic HRs. If possible, please consider modelling NLR/TILs as continuous variables or using pre‑specified cut‑offs, and clearly state in the Discussion that the current cut‑offs are exploratory and cohort‑specific.

-The cut-off value for NLR/TIL in ovarian cancer has not been established. Therefore, we used the cut-off value (4.23) determined by the ROC curve in this study. This aligns with the range (2.3–5.25) reported in previous studies on ovarian cancer (see discussion at line 357).

Zhang, Z.; Lang, J. The prognostic and clinical value of neutrophil-to-lymphocyte ratio (NLR) in ovarian cancer: a systematic review and meta-analysis. J. Med. Biochem. 2024, 43, 323–333.

2.2 Number of covariates vs events

There are 59 deaths for OS and 84 events for PFS, yet more than 10 covariates are included in the multivariable models. Please consider a more parsimonious model (fewer key covariates or penalised methods) and/or acknowledge the risk of over‑parameterisation as a limitation.

-As a limitation, we have specified that including more than ten covariates in the multivariate model carries the risk of over-parameterization (line 429).

2.3 Description of statistical tests

The Methods state that clinical/pathological data Mann–Whitney U tests, but Tables 2–3 compare categorical variables where chi‑square or Fisher’s exact tests would typically be used. Please specify which test was actually used for each table and re‑analyse if necessary.

-As you pointed out, we used the chi-squared test for the clinical/pathological data. We revised the description of the statistical analysis in Section 2.4 of the Methods and on lines 196, 241, and 295 of the Results.

2.4 Handling of missing lymph node status

Lymph node status is “Unknown” in 43/101 patients, but Cox models report only “Yes vs No.” Please clarify how “Unknown” was handled (exclusion, separate category, etc.) and its potential impact.

-In this study, "unknown" was treated as an exclusion. The significance of lymph node metastasis as a prognostic factor in advanced epithelial ovarian cancer is unclear. Additionally, since this study included a large number of cases with unknown lymph node status, those cases were excluded.

2.5 Confounding in the four‑group NLR × CD8+ iTIL analysis

Table 5 shows strong imbalances in chemotherapy regimen, albumin and ascites across the four groups (all p < 0.001), yet Figure 3 compares survival without adjustment. Please provide adjusted analyses including these covariates, or clearly discuss confounding as a limitation.

-As you pointed out, we have modified Figures 3(a-c) to include survival analyses adjusted for the covariates.

  1. Interpretation and discussion

3.1 “Independent prognostic factor”

Given the data‑driven cut‑offs, complex models and correlation issues, the term “independent prognostic factor” for NLR and CD8+ iTIL may be somewhat strong. I suggest softening the wording (e.g. “independent in this cohort”) and expanding the discussion of statistical limitations and the need for validation.

-This study found no correlation between TILs and NLR. However, as you instructed, we have softened the wording in the Discussion section to read ”independent prognostic factors in this cohort” (lines 346 and 402).

3.2 Choice of CD8+ iTIL as main TIL marker

Figure S3 shows the highest AUC for CD4+ cTIL, yet CD8+ iTIL is used as the central marker. Please briefly explain why CD8+ iTIL was selected as the primary TIL parameter (biological rationale, multivariable results, prior literature).

-As in previous reports, CD8+ iTILs were identified as independent prognostic factors. The multivariate analysis in this study also showed that CD8+ iTILs were prognostic factors and were selected as the primary TIL parameter (line 384-386).

3.3 Histology (HGSC vs non‑HGSC)

Figure S6 demonstrates higher CD8+/CD4+ iTIL density in HGSC vs non‑HGSC, while histology is not included as a covariate in the Cox models. Please consider including histology in multivariable analyses or provide HGSC‑only multivariable results, and highlight histology‑dependence of TIL cut‑offs as a limitation.

-Due to differences in biological characteristics among histological subtypes and the limited number of cases in each, this study performed a multivariate analysis without including histology. Setting NLR and TIL cutoff values for each subtype was also considered. However, the small number of cases per subtype made determining cutoff values from ROC curves difficult.

3.4 PD‑L1 (TPS) evaluation

PD‑L1 is dichotomised using TPS ≥ 1% on tumour cells only; the rationale and limitations are only briefly mentioned. A short justification for the 1% cut‑off and a note on not assessing immune‑cell PD‑L1 would strengthen the discussion.

-Based on the previous study, the cutoff value of TPS ≥ 1% was defined. In this study, only four cases had TPS ≥ 5%, which was a small number; therefore, a cutoff value greater than 1% was not selected.

Chen H,et al. PD-L1 Expression and CD8+ Tumor-infiltrating Lymphocytes in Different Types of Tubo-ovarian Carcinoma and Their Prognostic Value in High-grade Serous Carcinoma. Am J Surg Pathol. 2020 Aug;44(8):1050-1060.

  1. Minor presentation issues

4.1 “Number at risk” in Kaplan–Meier plots

Please add numbers at risk under all KM curves (Figures 1–3 and supplementary KM figures), which will improve interpretability.

-We have added numbers at risk under all KaplanMeier curves.

4.2 Consistency of terminology for chemotherapy regimens

Terms such as “Adjuvant,” “Neo‑adjuvant,” “Initial,” “First‑line,” and “NACT‑IDS” are used inconsistently across text and tables. Please define each in the Methods and harmonise terminology.

-As you pointed out, we have harmonised terminology (lines 86-89 and 102,103).

4.3 Abbreviations and legends

Abbreviations (c/i/sTILs, TPS, etc.) are not always defined in the figure legends or are used with minor variations. Please standardise abbreviations and ensure that each legend defines all terms used.

-We have added missing explanations for the abbreviations to the legends of each table and figure.

I hope these concise comments are helpful in revising the manuscript and strengthening the validity and clarity of your findings.

-Thanks to your review, our manuscript has significantly improved.

Round 2

Reviewer 2 Report

Comments and Suggestions for Authors

Most of my previous comments have been adequately addressed, and the manuscript (including the Supplement) has clearly improved. I thank the authors for their careful and thorough revisions. I now have only two remaining minor points before recommending acceptance.

1. Handling of “Unknown” lymph node status (Section 2.4 / Tables 4–6)
Table 1 and Table 6 indicate 43 patients with “Unknown” lymph node status, but the Methods and the footnotes of Tables 4–5 do not explain how these cases were handled in the Cox regression models (“Yes vs No” only). Please explicitly state in Section 2.4 and/or in the table footnotes how “Unknown” cases were treated in these analyses.

2. PD‑L1 (TPS) evaluation and limitations (Discussion)
Methods 2.3 clearly define PD‑L1 positivity as tumor cell TPS ≥1%, but the Discussion still does not briefly justify the choice of the 1% cut‑off or acknowledge that only tumor‑cell PD‑L1, not immune‑cell PD‑L1, was evaluated. As previously requested, I suggest adding a concise explanation of the rationale for the TPS ≥1% threshold (including the limited number of TPS ≥5% cases) and a sentence noting the limitation of assessing PD‑L1 only on tumor cells.

With these minor textual clarifications, I would be happy to recommend the manuscript for publication.

Author Response

Most of my previous comments have been adequately addressed, and the manuscript (including the Supplement) has clearly improved. I thank the authors for their careful and thorough revisions. I now have only two remaining minor points before recommending acceptance.

-We have made the necessary corrections regarding the two points you raised. Thank you for your thoughtful suggestions.

  1. Handling of “Unknown” lymph node status (Section 2.4 / Tables 4–6)
    Table 1 and Table 6 indicate 43 patients with “Unknown” lymph node status, but the Methods and the footnotes of Tables 4–5 do not explain how these cases were handled in the Cox regression models (“Yes vs No” only). Please explicitly state in Section 2.4 and/or in the table footnotes how “Unknown” cases were treated in these analyses.

-We added the following to the footnote of Tables 4-5: "Cases with 'unknown' lymph node status were excluded from the analysis."

  1. PD‑L1 (TPS) evaluation and limitations (Discussion)
    Methods 2.3 clearly define PD‑L1 positivity as tumor cell TPS ≥1%, but the Discussion still does not briefly justify the choice of the 1% cut‑off or acknowledge that only tumor‑cell PD‑L1, not immune‑cell PD‑L1, was evaluated. As previously requested, I suggest adding a concise explanation of the rationale for the TPS ≥1% threshold (including the limited number of TPS ≥5% cases) and a sentence noting the limitation of assessing PD‑L1 only on tumor cells.

-The reasons for setting the cutoff value to TPS 1% have been added to the Methods (lines 147,148) and the Discussion (lines 452,453).

When conducting quantitative assessments using image analysis software, it was necessary to combine multiple algorithms to identify PD-L1-positive cells of various sizes. However, due to the software's specifications, this is currently not possible. Evaluating CPS requires the aforementioned multiple algorithms. Conversely, TPS can be designed to focus solely on measuring PD-L1 in tumour cells. So, TPS was actually used for evaluation in this instance. The following sentence has been added: 'Taking into account the specifications of the HALO image analysis software, analysis was performed using TPS' (Methods, lines 146,147).